Effects of anesthetics pentobarbital sodium and chloral hydrate on urine proteome

Zhao Mindi 1
Li Xundou 1
Li Menglin 1
Gao Youhe 1 2 gaoyouhe@bnu.edu.cn
1 National Key Laboratory of Medical Molecular Biology, Department of Pathophysiology, Institute of Basic Medical Sciences, Chinese Academy of Medical Sciences, School of Basic Medicine, Peking Union Medical College , Beijing , China
2 Department of Biochemistry and Molecular Biology, Beijing Normal University, Gene Engineering and Biotechnology Beijing Key Laboratory , Beijing , China
Lomonte Bruno
Electronic publication date: 2015 Mar 12
Publication date: 2015
Volume: 3
Electronic Location ID: e813
Received 2015 Jan 11; Accepted 2015 Feb 13
Copyright: © 2015 Zhao et al.
Copyright year: 2015
Copyright holder: Zhao et al.
License: This is an open access article distributed under the terms of the Creative Commons Attribution License, which permits unrestricted use, distribution, reproduction and adaptation in any medium and for any purpose provided that it is properly attributed. For attribution, the original author(s), title, publication source (PeerJ) and either DOI or URL of the article must be cited.
License URL: https://creativecommons.org/licenses/by/4.0/

Keywords: Urine proteome, Anesthesia, Biomarkers

Funding: This work was supported by the National Basic Research Program of China (2012CB517606, 2013CB530805, 2014CBA02005 and 2013FY114100), Expertise-Introduction Project for Disciplinary Innovation of Universities (B08007). The funders had no role in study design, data collection and analysis, decision to publish, or preparation of the manuscript.

==============================
Urine can be a better source than blood for biomarker discovery since it accumulates many changes. The urine proteome is susceptible to many factors, including anesthesia. Pentobarbital sodium and chloral hydrate are commonly used anesthetics in animal experiments. This study demonstrated the effects of these two anesthetics on the rat urine proteome using liquid chromatography–tandem mass spectrometry (LC-MS/MS). With anesthesia, the urinary protein-to-creatinine ratio of all rats increased twofold. The relative abundance of 22 and 23 urinary proteins were changed with pentobarbital sodium or chloral hydrate anesthesia, respectively, as determined by label-free quantification. Among these changed proteins, fifteen had been considered as candidate biomarkers such as uromodulin, and sixteen had been considered stable in healthy human urine, which are more likely to be considered as potential biomarkers when changed, such as transferrin. The pattern of changed urinary proteins provides clues to the discovery of urinary proteins regulatory mechanisms. When determining a candidate biomarker, anesthetic-related effects can be excluded from future biomarker discovery studies. Since anesthetics take effects via nervous system, this study is the first to provide clues that the protein handling function of the kidney may possibly be regulated by the nervous system.

Introduction

Change is the most fundamental characteristic of biomarkers. Urine can be a better non-invasive source for biomarker discovery since it accumulates many changes (Gao, 2013). Changes introduced into the blood can be more sensitively detected in urine (Li, Zhao & Gao, 2014). As summarized in a recent paper (Gao, 2014b), in some previous biomarker studies, several potential biomarkers perform even better in urine than in blood (Huang et al., 2012; Payne et al., 2009; Wu et al., 2013). Urine proteome is affected by many factors such as age, gender, lifestyle and others. As a result, despite the advantage of urine as a better biomarker source, urine biomarker research can be difficult; changes in urine make sorting out factors directly associated with any particular condition much too complex, especially in human samples (Gao, 2013). Minimizing the confounding factors by using an animal model was illustrated in renal diseases (Gao, 2014c; Zhao et al., 2014). In fact, although the number of factors that can affect the urine proteome is still unknown, a better understanding of those factors’ effects on urine proteome can help to speed up biomarker discovery. It has been proposed that only changes of the stable components in urine proteome are more likely to become biomarkers (Sun et al., 2009). Other physiological factors such as water loading, sodium loading, cigarette smoking, diuretics and anticoagulants were found to change urine proteome as well (Airoldi et al., 2009; Li et al., 2014; Thongboonkerd et al., 2003).

The effects of medications on urine proteome also tend to be neglected when clinical experiments were designed. The patients-medicine, healthy-no medicine associations exist in all of the clinical biomarker studies. Therefore “pharmuromics,” which studies the effects of medicine on urine, was proposed (Gao, 2014a). Anesthetic is commonly used in animal experiments, as well as surgery. However, the effects of anesthetics on urine proteome are not usually considered. It is not clear whether anesthesia affects the urine proteome. In this study, the effects of pentobarbital sodium and chloral hydrate anesthesia on rat urine proteome were studied using liquid chromatography–tandem mass spectrometry (LC-MS/MS).

Materials and Methods

Experiment animals

Rats were purchased from the Institute of Laboratory Animal Science, Chinese Academy of Medical Science & Peking Union Medical College. The experiment was approved by Institute of Basic Medical Sciences Animal Ethics Committee, Peking Union Medical College (Animal Welfare Assurance Number: ACUC-A02-2013-015). All animals were kept with standard laboratory diet under controlled indoor temperature (22 ± 1 °C) and humidity (65–70%). The study was performed according to guidelines developed by Institutional Animal Care and Use Committee of Peking Union Medical College.

Rat models

Twelve male Sprague-Dawley rats (weight = 200 g) were divided into two groups. One group was anesthetized by intraperitoneal injection of pentobarbital sodium (n = 6, 50 mg/kg), and the other group was by chloral hydrate (n = 6, 300 mg/kg). Urine samples before anesthesia were collected as control (about 2 mL). Anesthesia affected urine was collected for three hours during anesthesia (about 2 mL). Anesthesia was supported for four hours and the activities of the anesthetics were detected by measuring muscle relaxation. The urinary protein and creatinine concentration were measured at the Peking Union Medical College Hospital. The self-controlled experiment was conducted in two phases: for the discovery phase, differential protein identification was performed in three independent rats each group; for the validation phase, samples were obtained from the three remaining rats.

Sample preparation

Urine was centrifuged at 2,000 g for 30 min immediately after collection. Three volumes of acetone were added after removing the pellets and precipitated at 4 °C. Then, lysis buffer (8 M urea, 2 M thiourea, 25 mM dithiothreitol and 50 mM Tris) was used to re-dissolve the pellets. Proteins were digested by trypsin (Trypsin Gold, Mass Spec Grade, Promega, Fitchburg, Wisconsin, USA) using filter-aided sample preparation methods (Wisniewski et al., 2009). Briefly, after proteins were loaded on the filter unit (Pall, Port Washington, New York, USA), UA buffer (8 M urea in 0.1 M Tris–HCl, pH 8.5) and 50 mM NH4HCO3 was added. Proteins were denatured at 50 °C for 1 h by the addition of 20 mM dithiothreitol and alkylated in the dark for 40 min by the addition of 50 mM iodoacetamide. Proteins were digested by trypsin (1:50) at 37 °C overnight. The digested peptides were desalted using Oasis HLB cartridges (Waters, Milford, Massachusetts, USA).

LC-MS/MS analysis

The digested peptides were dissolved in 0.1% formic acid and loaded on a Michrom Peptide Captrap column (MW 0.5–50 kD, 0.5 × 2 mm; Michrom Bioresources, Auburn, California, USA). The eluent was transferred to a reversed-phase microcapillary column (0.1 × 150 mm, packed with Magic C18, 3 µm, 200 Å; Michrom Bioresources, Auburn, California, USA) by an Agilent 1200 HPLC system. Peptides were analyzed by a LTQ-OrbitrapVelos mass spectrometer (Thermo Fisher Scientific, Bremen, Germany). The LTQ-OrbitrapVelos was operated in data-dependent acquisition mode. Survey MS scans were acquired in the Orbitrap using a 300–2,000 m/z range with the resolution set to 60,000. The 20 most intense ions per survey scan were selected for collision-induced dissociation fragmentation, and the resulting fragments were analyzed in the LTQ. Dynamic exclusion was employed with a 60 s window to prevent the repetitive selection of the same peptide.

Data analysis

All MS/MS spectra were analyzed using the Mascot search engine (version 2.4.1, Matrix Science, London, UK), and proteins were identified by searching against the Swissprot_ 2013_ 07 database (taxonomy: Rattus; containing 9,354 sequences). The parameters were set as follows: carbamidomethylation of cysteines was set as a fixed modification, and oxidation of methionine and protein N-terminal acetylation were set as variable modifications. Trypsin was set as the digestion enzyme, and two missed trypsin cleavage sites were allowed. The precursor mass tolerance was set to 10 ppm, and the fragment mass tolerance was set to 0.5 Da. Peptide and protein identifications were validated by Scaffold (version 4.0.1, Proteome Software Inc., Portland, Oregon, USA). Peptide identifications were accepted if they could be detected with ≥95.0% probability by the Scaffold local false discovery rate algorithm, and protein identifications were accepted if they could be detected with ≥99.0% probability and contained at least 2 identified peptides (Nesvizhskii et al., 2003). The acquired raw files were loaded to Progenesis LC-MS/MS software (version 4.1, Nonlinear, Newcastle upon Tyne, UK), and label-free quantification was conducted as previously described (Hauck et al., 2010). For quantification, all peptides (with Mascot score >30 and p < 0.01) of an identified protein were included.

Western blot analysis

Urine proteins were prepared as described in Materials and Methods; 20 µg of each sample were separated by 10% SDS-PAGE and transferred to PVDF membranes (Whatman, Maidstone, UK) in transfer buffer (10% methanol, 25 mM Tris base, 192 mM glycine, PH 8.0). Membranes were incubated overnight at 4 °C with primary antibody against alpha-1-antiproteinase (Species reactivity: rat; dilution 1:1000; ab106582, Abcam, Cambridge, UK) or transferrin (Species reactivity: rat; dilution 1:10000; ab82411, Abcam, Cambridge, UK). The membranes were then washed and incubated with peroxidase-conjugated anti-chicken and anti-rabbit IgG (1:10000; zSgb-Bio, Beijing, China) at room temperature for 2 h and proteins were visualized using enhanced chemiluminescence (ECL) reagents. Intensity of each protein band was quantified using Image J analysis software (National Institutes of Health, Bethesda, Maryland, USA).

Results

Urine protein-to-creatinine ratios were increased with either pentobarbital sodium or chloral hydrate anesthesia

The urinary creatinine concentrations were first reduced with anesthesia than in normal condition (in pentobarbital sodium group, creatinine decreased to 0.9-fold and in chloral hydrate group, also decreased to about 0.9-fold; Table S1). When compared with changes of creatinine concentration, changes of urinary proteins concentrations were much more significant. As a result, the urine protein-to-creatinine values with anesthesia increased 2.4-fold (in pentobarbital sodium group, 107.1 ± 21.1 versus 259.1 ± 81.1 mg/mmol, n = 6, P value <0.05) and 2.1-fold (in chloral hydrate group, 107.5 ± 16.5 versus 220.8 ± 79.0 mg/mmol, n = 6, P value <0.05). With pentobarbital sodium and chloral hydrate anesthesia, the urine protein-to-creatinine ratio of all rats were significantly increased in both groups, which were consistent with the values that have been reported in previous studies (Mercatello et al., 1991; Vaden et al., 2010). Figure 1 showed the different effects of each anesthetic on rat urine protein concentration.

Figure 1 Urine protein-to-creatinine ratios before and after anesthesia (n = 6 each group).

Table 1 Changes in the urine proteome identified by LC-MS/MS with two anesthetics.

			Pentobarbital sodium group fold change	Chloral hydrate group fold change		
Accession	Description	P value	Rat 1	Rat 2	Rat 3	Rat 7	Rat 8	Rat 9	Candidate biomarkers	
P17475	Alpha-1-antiproteinase	0.034	6.1↑	5.6↑	8.5↑	3.4↑	8↑	3.3↑	Yes	
P07154	Cathepsin L1	0.003	2.4↓	3.2↓	4↓	2.4↓	2↓	8.9↓	Yes	
P07522	Pro-epidermal growth factor	0.001	2.5↓	2.8↓	3.4↓	5.4↓	3.3↓	2.1↓	Yes	
P00758	Kallikrein-1	0.002	2.1↓	3.3↓	3.3↓	7.5↓	2.7↓	2↓	No	
Q5XI43	Matrix-remodeling-associated protein 8	0.006	2.8↓	3.1↓	2.6↓	9.3↓	5.3↓	2.8↓	No	
P15083	Polymeric immunoglobulin receptor	0.020	2.6↓	2.3↓	2.2↓	3↓	2.7↓	2.8↓	No	
P27590	Uromodulin	0.006	3↓	5.2↓	7↓	3.7↓	2↓	2.2↓	Yes	
P02770	Serum albumin	0.042	5.5↑	3.1↑	5.4↑	–	–	–	Yes	
P12346	Serotransferrin	0.049	6.8↑	2.1↑	4.3↑	–	–	–	Yes	
P32038	Complement factor D	0.046	2.2↑	2.4↑	3.9↑	–	–	–	No	
P10959	Carboxylesterase 1C	0.034	3.5↑	3.9↑	4.6↑	–	–	–	No	
P20761	Ig gamma-2B chain C region	0.030	7.2↑	3.3↑	9.1↑	–	–	–	No	
P50123	Glutamyl aminopeptidase	0.044	2.1↓	2.5↓	2.1↓	–	–	–	No	
Q62867	Gamma-glutamyl hydrolase	0.046	2.2↓	3↓	3.3↓	–	–	–	Yes	
P15684	Aminopeptidase N	0.039	2.4↓	4.4↓	5.7↓	–	–	–	Yes	
P26051	CD44 antigen	0.006	2.9↓	2.5↓	2.6↓	–	–	–	No	
P36373	Glandular kallikrein-7, submandibular/renal	0.021	2.1↓	2.2↓	3.5↓	–	–	–	Yes	
P98158	Low-density lipoprotein receptor-related protein 2	0.004	2.1↓	3.6↓	2.1↓	–	–	–	No	
Q64230	Meprin A subunit alpha	0.000	2.7↓	3.6↓	3.4↓	–	–	–	Yes	
P28826	Meprin A subunit beta	0.031	3.5↓	4.9↓	10.9↓	–	–	–	No	
Q64319	Neutral and basic amino acid transport protein rBAT	0.014	2.5↓	2.7↓	4.5↓	–	–	–	Yes	
P29598	Urokinase-type plasminogen activator	0.048	2.5↓	2.2↓	2.7↓	–	–	–	No	
Q6DGG1	Alpha/beta hydrolase domain-containing protein 14B	0.004	–	–	–	3.4↑	8↑	3.3↑	No	
Q6IRK9	Carboxypeptidase Q	0.037	–	–	–	2.9↑	3.6↑	4.8↑	No	
P08649	Complement C4	0.028	–	–	–	11.2↑	3.1↑	14↑	No	
P61972	Nuclear transport factor 2	0.026	–	–	–	3.1↑	3.1↑	6.5↑	No	
P02625	Parvalbumin alpha	0.047	–	–	–	5.9↑	5↑	12.5↑	Yes	
Q920A6	Retinoid-inducible serine
carboxypeptidase	0.019	–	–	–	4.2↑	4.2↑	4.7↑	No	
P82450	Sialate O-acetylesterase	0.016	–	–	–	5↑	9.4↑	2.4↑	No	
P07632	Superoxide dismutase [Cu-Zn]	0.019	–	–	–	2.6↑	4.9↑	3.1↑	Yes	
P02650	Apolipoprotein E	0.032	–	–	–	2↓	5.2↓	3.3↓	No	
Q9R0T4	Cadherin-1	0.039	–	–	–	2.7↓	3.5↓	2.1↓	Yes	
P31211	Corticosteroid-binding globulin	0.038	–	–	–	3.6↓	2.1↓	2.8↓	No	
Q9JJ40	Na(+)/H(+) exchange regulatory
cofactor NHE-RF3	0.047	–	–	–	3.0↓	2.7↓	3.2↓	Yes	
P08460	Nidogen-1 (Fragment)	0.020	–	–	–	4↓	2.5↓	4.1↓	No	
Q63083	Nucleobindin-1	0.043	–	–	–	16.7↓	10.8↓	3.7↓	No	
P83121	Urinary protein 3	0.033	–	–	–	2.2↓	3.3↓	2.1↓	No	
P05371	Clusterin	0.040	–	–	–	5.7↓	5.8↓	2.2↓	No	
Notes.

“–” means no significant changes (fold changes >2 in all three samples).

Urinary proteome changes with anesthesia

Twelve urine samples before and after anesthesia from 6 rats (n = 3 in each group) in the pentobarbital sodium and chloral hydrate group were profiled by LC-MS/MS. In the pentobarbital sodium and chloral hydrate group, label-free quantitation data of proteins identified were listed in Table S2.

In the pentobarbital sodium group, the relative abundance of 22 proteins changed according to the following criteria: fold change >2 for each rat and p value <0.05 (data were analyzed by t test); 6 proteins had increased relative abundance and 16 proteins had decreased relative abundance. In the chloral hydrate group, the relative abundance of 23 proteins changed: 9 proteins had increased relative abundance and 14 proteins had decreased relative abundance. Among the proteins with altered relative abundance, 7 had the same trends in all six rats that were anesthetized with either pentobarbital sodium or chloral hydrate; one protein increased relative abundance and six proteins had decreased relative abundance (Table 1).

Verification of affected proteins by Western blot

Two changed proteins were selected to be validated in six more rats for the following reasons: (1) were identified previously in biomarker discovery; (2) were at relatively high abundance and easier to be detected in western blot; (3) had commercially available antibodies. In the pentobarbital sodium group, the levels of transferrin were analyzed and in the chloral hydrate group, the levels of alpha-1-antiproteinase were analyzed. With anesthesia, transferrin and alpha-1-antiproteinase expression levels were upregulated in three more rats (Fig. 2), consistent with the MS quantification data.

Figure 2 Semi-quantitative western blot analysis of two proteins.

(A) Levels of urinary transferrin before and after pentobarbital sodium anesthesia. (B) Levels of urinary alpha-1-antiproteinase before and after chloral hydrate anesthesia. (C) Quantitation of the transferrin by western blot analysis from 3 independent biological replicates. (D) Quantitation of the alpha-1-antiproteinase by western blot analysis from 3 independent biological replicates * indicates p < 0.05 (data were analyzed by t test).

Comparison with previous studies

In the pentobarbital sodium anesthesia group, the relative abundance of 22 proteins were changed. Compared with the Urinary Protein Biomarkers Database (Shao et al., 2011), 11 out of 22 proteins were considered as candidate biomarkers, such as uromodulin and serotransferrin. Among these proteins, some exhibited the opposite trend. For example, the relative abundance of aminopeptidase N was increased in septic rats with acute renal failure (Wang et al., 2008), whereas their relative abundance decreased with pentobarbital sodium anesthesia. In the chloral hydrate anesthesia group, the relative abundance of 23 proteins changed and chloral hydrate had a relatively different impact on the urine proteome. Compared with the Urinary Protein Biomarkers Database, 8 out of 23 proteins were considered as candidate biomarkers, such as uromodulin and parvalbumin alpha. However, the relative abundance of clusterin was increased under conditions of gentamicin administration (Takahashi, 1995), but it decreased with chloral hydrate anesthesia.

Rat proteins were converted to their human orthologs using the Ensembl homolog database as reported (Jia et al., 2013). Stable proteins in healthy human urine are more likely to become candidate biomarkers when changed (Sun et al., 2009). In this study, differently expressed proteins with anesthesia were compared with the human core urinary proteome, which were considered stable and relatively high in abundance. Data from the “stable urinary proteome,” which represented the common and most easily identifiable proteins from urine, were determined by Mann (Nagaraj & Mann, 2011). The dataset contains 587 proteins that were identified in each of the 7 participant’s urinary proteomes on three consecutive days. The changes of high abundant proteins are likely to be real, as it is unlikely to be caused by data dependent sampling of low abundant peptides by MS. 6 out of 22 proteins (Uromodulin, Kallikrein-1, Serotransferrin, Serum albumin, Gamma-glutamyl hydrolase, Neutral and basic amino acid transport protein rBAT) affected by pentobarbital sodium had stable relative abundance in healthy human urine. Twelve out of 23 proteins (Uromodulin, Kallikrein-1, Superoxide dismutase (Cu–Zn), Putative uncharacterized protein, Parvalbumin alpha, Corticosteroid-binding globulin, E-cadherin, Alpha/beta hydrolase domain-containing protein 14B, Retinoid-inducible serine carboxypeptidase, Apolipoprotein E, Na(+)/H(+) exchange regulatory cofactor NHE-RF3, Nucleobindin-1) affected by chloral hydrate were stable. Two proteins (Uromodulin, Kallikrein-1) were shared by both groups (Table 2 listed the changed proteins which exist in human core urinary proteins).

Table 2 Changed proteins with anesthesia which exist in human stable urinary proteome and their corresponding human orthologs.

Group	Uniprot (rat)	Human ensembl
gene ID	Uniprot (human)	Protein name	Related-disease	
Both group	P27590	ENSG00000169344	P07911	Uromodulin	Fanconi Syndrome (Cutillas et al., 2004)	
	P00758	ENSG00000167748	P06870	Kallikrein-1	None	
Pentobarbital sodium group	Q64319	ENSG00000091513	P02787	Serotransferrin	Diabetic Nephropathy (Narita et al., 2004)	
	Q628 67	ENSG00000163631	P02768	Serum albumin	Nephrotoxicity (Nordberg et al., 2005)	
	P12346	ENSG00000137563	Q92820	Gamma-glutamyl hydrolase	Uranium Nephrotoxicity (Malard et al., 2009)	
	P02770	ENSG00000138079	Q07837	Neutral and basic amino acid transport protein rBAT	Sodium Loading (Thongboonkerd et al., 2003)	
Chloral hydrate group	P07632	ENSG00000142168	P00441	Superoxide dismutase (Cu–Zn)	Nephritis (Curtis et al., 1989)	
	Q6IRK9	ENSG00000104324	Q9Y646	Putative uncharacterized protein	None	
	P02625	ENSG00000100362	P20472	Parvalbumin alpha	Skeletal Muscle Toxicity (Dare et al., 2002)	
	P31211	ENSG00000170099	P08185	Corticosteroid-binding globulin	None	
	Q9R0T4	ENSG00000039068	P12830	E-cadherin	Diabetic Nephropathy (Jiang et al., 2009)	
	Q6DGG1	ENSG00000114779	Q96IU4	Alpha/beta hydrolase domain-containing protein 14B	None	
	Q920A6	ENSG00000121064	Q9HB40	Retinoid-inducible serine carboxypeptidase	None	
	P02650	ENSG00000130203	P02649	Apolipoprotein E	Bladder Cancer (Linden et al., 2012)	
	Q9JJ40	ENSG00000174827	Q5T2W1	Na(+)/H(+) exchange regulatory cofactor NHE-RF3	Aldosteronism (van der Lubbe et al., 2012)	
	Q63083	ENSG00000104805	Q02818	Nucleobindin-1	None	

Discussion

Two validated changing proteins, transferrin and the alpha-1-antiproteinase, are two of the most common markers of renal diseases. Transferrin is a plasma protein that transports iron through different tissues and organs (Crichton & Charloteaux-Wauters, 1987). The blood transferrin is used to determine the cause of anemia and examine iron metabolism. Urinary transferrin is upregulated in many diseases such as diabetic nephropathy, IgA nephropathy, ureteropelvic junction obstruction and bladder cancer (Shao et al., 2011). Alpha-1-antiproteinase can inhibit many proteases thus protects tissues from enzymes of inflammatory cells (Wu & Foreman, 1991). Alpha-1-antiproteinase is also upregulated in many diseases such as kidney calculi, nephrotic syndrome, bladder cancer and focal segmental glomerulosclerosis (Shao et al., 2011). As these two candidate biomarkers are affected by anesthetics like pentobarbital sodium or chloral hydrate, it is necessary to exclude anesthetic related effects in future biomarker discovery studies.

Seven changed proteins shared the same trend in both groups, which could be explained by the common mechanisms of action of two general anesthetics. Pentobarbital sodium at anesthetic dose inhibits Ca2+-dependent release of neurotransmitters and increases the duration of Cl− channel opening at the GABAA receptor (Orser et al., 1998; Pistis et al., 1999). Chloral hydrate also potentiates GABA-activated Cl− current in central nervous system neurons by its main active metabolite trichloroethanol (Peoples & Weight, 1994). The common effects of these two anesthetics on urine proteome suggest that the nervous system is possibly involved in regulation of urinary proteins. But exactly how these two anesthetics affect urinary proteins remains unknown. It may include direct and/or indirect effects on renal functions.

Central GABA receptor stimulation reduces renal sympathetic nerve discharge (Antonaccio & Taylor, 1977), which induce vasodilatation, especially in the arcuate and interlobular arteries (Kirchheim et al., 1987). Central administration of GABA agonists reduce blood pressure and heart rate (Antonaccio, Kerwin & Taylor, 1978), which could affect renal blood flow, glomerular filtration rate, renal tubular reabsorption rate (Holstein-Rathlou, Christensen & Leyssac, 1982; Mercatello, 1990) and possibly urinary proteins.

It was proposed that GABA antagonizes the central effects of renin (Abe et al., 1988). The lower release of renin may consequently affect the renal sodium metabolism (Zacchia & Capasso, 2008), which may explain why Na (+)/H (+) exchange regulatory cofactor and parvalbumin (a key protein in early distal tubule Na+ reabsorption) were affected with chloral hydrate anesthesia.

The fact that changed proteins with pentobarbital sodium and chloral hydrate anesthesia were not all the same suggested that the two anesthetics might have differences in the modes of action. Chloral hydrate also targets on the 5-HT3 receptor (Bentley & Barnes, 1998), which may help to explain the different effects of the two anesthetics. Previous study also showed that pentobarbital sodium anesthesia may influence hematologic values such as clotting time and partial thromboplastin time (Gentry & Black, 1976), which may explain why the kallikrein-1 and urokinase-type plasminogen activator changes with pentobarbital sodium anesthesia.

Whether these changes are influenced by anesthetic dose, the depth of the anesthesia and the time of administration require further study. The pattern of urinary proteins with altered relative abundances makes it possible to exclude interferences from anesthetics in future biomarker discovery studies. Besides, the analysis above suggests that urinary proteins may be able to reflect the functional changes as far as central nerve system. A better understanding of this mechanism will help to understand renal physiology, pathophysiology and the relationship between biomarkers and related diseases.

Supplemental Information

Table S1 Urinary creatinine concentrations of all samples

Click here for additional data file.

Table S2 Label-free quantitation data of proteins identified in both anesthesia

(A) Label-free quantitation data of proteins identified in pentobarbital sodium group. (B) Label-free quantitation data of proteins identified in chloral hydrate group.

Click here for additional data file.

Additional Information and Declarations

Competing Interests

Author Contributions

Field Study Permissions

Data Deposition

The authors declare there are no competing interests.

Mindi Zhao performed the experiments, analyzed the data, contributed reagents/materials/analysis tools, wrote the paper, prepared figures and/or tables.

Xundou Li and Menglin Li contributed reagents/materials/analysis tools.

Youhe Gao conceived and designed the experiments, reviewed drafts of the paper.

The following information was supplied relating to field study approvals (i.e., approving body and any reference numbers):

The experiment was approved by Institute of Basic Medical Sciences Animal Ethics Committee, Peking Union Medical College (Animal Welfare Assurance Number: ACUC-A02-2013-015).

The following information was supplied regarding the deposition of related data:

All raw data identified by mass spectrometry are publicly available at and .

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
