# Peer review of "Effects of anesthetics pentobarbital sodium and chloral hydrate on urine proteome"

_PeerJ, doi:10.7717/peerj.813_

## Round 0.1 · original submission · Minor Revisions

Zhao and Gao report in this manuscript the influence of two common anesthetics (pentobarbital sodium and chloral) on the urine proteome using a rat model. The study aims to provide a basis for the selection of useful biomarkers in human disease. The findings also help to exclude anesthetic-related effects in future biomarker discovery studies based on urinary proteins. The study design is clear and the label-free LC-MS/MS methods adequate and representative, supporting the main findings and conclusions. A possible link between nervous system physiology and renal effects is suggested by the findings in rats, although the possibility of direct effects of anesthetics on renal physiology cannot be excluded. The authors clearly ackowledge this.

Overall the manuscript is well written and presented (although few suggestions/observations are given below). It adds valuable new knowledge, and should be of interest in the field. Therefore, a revised version addressing the points raised by reviewers (especially point#2 of the reviewer on the protein/creatinine ratios - validity of findings) could be acceptable for publication in PeerJ.

Minor points to be considered to improve presentation are:

1.Line 24: avoid repeating "animal model" in the sentence.

2.Line 31: this sentence needs to be improved for clarity.

3.Line 33: "purposed...", do you mean "proposed"?

4.Line 51: "was collected during anesthesia....". The description should be more specific regarding times.

5.Line 74: spell out "collision-induced dissociation" for CID

6.Line 86: spell out "false discovery rate" for FDR

7.Line 98: the description of the antibodies used for western blot experiments needs to be improved, please indicate the animal species origin for the primary antibodies, and the specificity of the secondary, conjugated antibodies.

8.Statistics: throughout the manuscript, no indication of the type of test used to establish statistical significance of differences is provided. Please clarify.

9.Line 113: "were individually identified by...", a more adequate term could be "profiled by...".

10.Line 171: anesthetics

11.Line 178: "or indirect effects..." would be more clear to change to "....or indirect effects on renal functions"

12.Line 184: antagonizes
13.Line 184: "the less release" could be changed to "the lower release"

14.Table 1. The exact meaning of the symbol "____" should be indicated in a footnote.

15.Figs 1 and 2: indicate statistical test used.

Reviewer 1 ·

Basic reporting

The reviewed paper presents the study on the influence of two anesthetics on the urinary proteom of examined rats. The main goal of the study was to reveal potential confounders related to anesthetics when analysing the urinary biomarkers. Interesting issue.

The some small correction should be done in the text. in line 33 the is "purposed" instead of proposed ? In line 159 should be 'are two of the most', in line 171 missed "s" in anesthesia.

Experimental design

Some question/comments rise with regards to methodology.

1. Three subjects from each compared group and their pairwise controls are rather narrow study. It would be valuable to extend the number of subjects, Chosen biomarkers should be verified by SRM/MRM, more accurate than label-free method. However, the preliminary study are to accept, but required explanation of applied method with known limiations.

2.The urine was collected during anesthesia. It would be valuable to examine the protein excretion changes in the time course, with no limitation to only urine collected during anesthesia. This is rather a hypothetical case when urine sample is urgently collected on the surgical table, anyway processed for time consuming MS/MS assays. It would be valuable to complete the studies on later times of collection to capture how long, if really, the effect of anesthesia may confound the proteom. If the study is preliminary, this issue should be commented.

3. How long anesthesia was supported and what was urine volumes obtained from rats ? What was the method of creatinine concentration assay ? Those information should be included in the paper.

Validity of the findings

1. See point.1.

2. Althoug two protein excretion trends were semi-quantitatively proved by Western-blott, it should be proven that the urinary protein-creatinine ratios deterimned by mS/MS were changed really due to protein change rather than creatinine concentration. In fact, both are changed, but what were creatinine concentrations before and after anesthesia ? The measured creatinine concentations should be included in the paper. If the observed results reflects immediate changes of protein excretion during anesthesia, one can expect also changes in creatinine, especially that muscle relaxation is the effect of anesthetics action. This requires data completion and a comment.

---

## Round 0.2 · accepted · Accept

Clarification on the inclusion of two additional authors is satisfactory.

Amendments to the original submission are acceptable, and improve the clarity of this communication.